# Study on Enhanced Methods for Calculating NH_3_ Emissions from Fertilizer Application in Agriculture Sector

**DOI:** 10.3390/ijerph182111551

**Published:** 2021-11-03

**Authors:** Jiyun Woo, Saenun Song, Seongmin Kang, Eui-Chan Jeon

**Affiliations:** 1Department of Climate and Environment, Sejong University, Seoul 05006, Korea; woojune92@gmail.com (J.W.); ioiyz@naver.com (S.S.); 2Climate Change & Environment Research Center, Sejong University, Seoul 05006, Korea; smkang9804@gmail.com

**Keywords:** agriculture, fertilizer application, ammonia, particulate matter, inventory

## Abstract

Ammonia is a representative PM-2.5 secondary product, and the need for management is emerging as health and living damage caused by fine particulate matter worsens. The main source of ammonia is the agricultural sector, and in Korea, 79% of the total ammonia emissions are emitted from the agricultural sector. Among them, there is high uncertainty about how to calculate emissions from ammonia discharged from fertilizer use, and inventory in the U.S. and Europe is borrowed, so inventory needs to be improved according to the situation in Korea. In this study, the ammonia inventory in the agricultural sector in Korea and abroad was examined, and additional activity data that can be used were reviewed. In addition, in order to improve the emission calculation method, the emission was calculated in three ways by different factors. As a result, it was confirmed that the amount of discharge varies depending on the type of soil use or whether cultivated crops are considered, and the possibility of excessive fertilizer top-dress by farmers was confirmed. In order to calculate the emission at a more detailed level based on this study, basic data such as fertilizer input method and regional distribution of crops should be systematically collected, and related follow-up studies should be conducted.

## 1. Introduction

Various studies worldwide have reported damages to human health caused by particulate matter, which has become a critical social issue [1,2,3]. Particles smaller than or equal to 2.5μm in diameter are defined as fine particulate matter (PM-2.5). Since the diameter of PM-2.5 is less than 1/20 the diameter of human hair, these particles can reach pulmonary alveoli or bronchial tubes without being filtered by the nose and cause respiratory diseases. Additionally, they may affect various other organs and cause cardiovascular disease, lung cancer, cell aging, and infections [4,5,6,7]. Hence, the International Agency for Research on Cancer, an affiliated organization of the World Health Organization, specified particulate matter as a carcinogen in 2013 [8].

Depending on their formation, PM-2.5 are categorized as either primary or secondary PM-2.5. Primary PM-2.5 are generated from the source and are directly emitted as solid fine dust. Secondary PM-2.5 refers to emitted SOx and NOx that form ammonium sulfate or ammonium nitrate by reacting with substances such as ammonia (NH_3_) in the air [9,10,11]. Secondary PM-2.5 accounts for over 72% of PM-2.5 generated in South Korea [12], and their importance is increasing due to an increase in the number of high PM-2.5 concentration days.

NH_3_ concentrations are closely related to the change in PM-2.5 concentration, thus requiring accurate emission calculations and the identification of emission sources. However, in comparison to SOx and NOx, NH_3_ is under less strict control in South Korea.

A management system is enforced for the total SOx and NOx load, and these pollutants are measured in real time at large-scale workplaces; currently, there is no such management scheme for NH_3_.

According to South Korea’s national emission statistics in 2017, total NH3 emissions were 308,298 tons, of which the majority (79.3%) was generated from the agricultural sector [13]. Chemical fertilizers are excessively applied in South Korean farmlands to increase productivity. The aggregate amount of chemical fertilizer application in South Korea ranked the highest among Organisation for Economic Co-operation and Development countries [14]. Therefore, regulation is necessary for NH3 emissions, which are generated from volatilization from soils due to excessive chemical fertilizer application.

Currently, the list of emission sources and most emission factors are adopted from overseas systems for calculating NH3 emissions of fertilized farmland in South Korea. In addition, an accurate emission calculation method based on the agricultural environment in South Korea is required because the calculated emissions are evenly distributed for each month and fail to reflect the circumstances of the actual environment.

In this study, global NH3 emission calculation methods were reviewed to devise potential enhancements for an accurate and a reliable method of NH3 emission calculation in the South Korean agricultural sector.

## 2. Review of the Ammonia Measurements Methods in Agriculture Sector

Each country uses different methods to control and account for NH_3_ emission sources in the agricultural sector, taking country-specific factors into account. The United States (US) and the European Union (EU) provide detailed guidelines for calculating NH_3_ emissions in the agricultural sector. These guidelines are reviewed in Table 1.

In the US, agricultural emissions are classified into four categories: crop cultivation, dust from livestock, fertilizer application, and livestock feces management. NH_3_ emission sources from fertilizer application are classified into 14 nitrogen-based fertilizers (including anhydrous ammonia, aqueous ammonia, and urea). The Fertilizer Emission Scenario Tool for CMAQ is used in the emission calculations, and the soil profile, climate variables, volume of fertilizer supply, and the status of the cultivation area are used as activity data. Information on cultivation management (fertilizer application period, fertilizer components, application methods, and application quantities) is obtained through surveys, and the soil nutrient content and crop nutrient demand are estimated for their use in the NH_3_ emission calculation. The emission factors are calculated monthly for each administrative region, considering the percentage of NH_3_ emissions due to the total nitrogen fertilizer use. The calculated emissions consider the crop cultivation circumstances in the US [15].

The EU controls NH_3_ emissions through country-specific methods stated in the EMEP/EEA Air Pollutant Emission Inventory Guidebook. The guidebook classifies the agricultural sector into four emission sources, according to the NFR (Nomenclature For Reporting) code: manure management, crop production, agricultural soils, and other agriculture, which includes the use of pesticides and the burning of agricultural waste in the field. NH_3_ emission sources from fertilizer applications are restricted to those from nitrogen-based fertilizers. There are three emission calculation methods: Tier 1, Tier 2, and Tier 3. In Tier 1, the basic emission factors are multiplied by the annual fertilizer supply volume. In Tier 2, the climatic zone and soil pH are considered based on the 2006 Intergovernmental Panel on Climate Change Guidelines for National Greenhouse Gas Inventories and the unique emission factors for 11 nitrogen-based fertilizers are used. Finally, specific data such as crop growth processes, land type, climate conditions, the local distribution by crop type, and average nitrogen demand by crop type are utilized in Tier 3 for greater precision [16].

In South Korea, the agricultural sector is classified into two emission sources: fertilized farmland and feces management. The emission sources within fertilized farmland are further classified into 10 types of nitrogen-based fertilizers (such as urea, compound fertilizers, and ammonium sulfate). The emissions are calculated by multiplying the regional fertilizer supply by the fertilizer nitrogen content and the emission factor. Then, the emission quantities are allocated according to the farmland area. The volume of fertilizer supplied regionally by the National Agricultural Cooperative Federation is used, since accurate figures of the applied fertilizer quantities are difficult to obtain. In addition, the agriculturally active period is assumed to be eight months (from March to October), considering the average temperatures. During this period, NH_3_ emissions are evenly distributed. However, the exact fertilizer type and application location are unknown, and the assumption that the annual supply of fertilizer is completely consumed in the supplied region differs from reality. Lastly, the even distribution of NH_3_ emissions from March to October does not reflect actual agricultural activities, and emissions from greenhouses are not considered [17].

The US and most EU member countries use models to calculate national emissions. This allows for the exact location and the management of specific emissions information (cultivation plans, fertilizer application period, crop nutrient demand, etc.) to be taken into account. Currently, obtaining specific information related to NH_3_ emissions and model applications are difficult in South Korea. Nevertheless, accuracy in emission control can be improved using the acquired data on farmland area, cultivated crops, and nationally recommended cultivation methods by crop type.

## 3. Materials and Methods

### 3.1. Study Site Selection

Jeolla Province was selected as the pilot study area for the development of an enhanced NH_3_ emission calculation method based on fertilized farmland (Figure 1). This region emits the largest amount of NH_3_ from the largest area of arable land in South Korea. National emissions statistics in 2017 revealed that fertilized farmland emitted 17,754 tons of NH_3_ out of 244,335 tons of total NH_3_ emissions, and 6189 tons (34.9%) originated from Jeolla Province. South Korea has a total farmland area of 1,320,795 ha, and 493,059 ha (30.4%) area is distributed in Jeolla Province [18].

### 3.2. Fertilized Farmland NH_3_ Emission Calculation Method

In South Korea, NH_3_ emissions are calculated based only on the regional fertilizer supply volume and the nitrogen content. They are allocated to administrative districts based on the farmland area during the agriculturally active period (March–October), which does not reflect the actual duration or location of fertilization. The spatial and temporal resolution of the NH_3_ emission calculation method was enhanced and compared to the existing method as shown in Table 2. In this study, 2017 data was used for comparison with the latest national emissions (2017). In the enhanced calculation method, the most common land use types in South Korea (rice paddies, fields, greenhouses, and pomiculture), according to their current emission source classification system depending on the current fertilization types, the crops cultivated by region and farmland type, the standard fertilizer application amount for each crop type, and farming schedules, were considered.

#### 3.2.1. NH_3_ Emission Calculation Method Considering the Volume of Fertilizer Supply

Method I (NH_3_ emission calculation considering the volume of supply for each fertilizer type) considers the volume of supply for each fertilizer type to calculate NH_3_ emission. This method is currently used in NH_3_ emission calculations from fertilization applications in South Korea. The emission sources’ classification system is based on fertilizer types (including urea, compound fertilizers, and ammonium sulfate) as shown in Table 3.

The NH_3_ emission is calculated by multiplying the volume of fertilizer and nitrogen content by the fertilizer type and emission factor as shown in Equation (1). The EPA emission factors were adopted for ammonium sulfate and other nitrogen-based fertilizers, and emission factors were developed in South Korea and are used for urea and compound fertilizers as shown in Table 4.
(1)E=A×N/100×EF
where *E* is NH_3_ Emission (kg-NH_3_/yr); *A* is supply volume of fertilizer (ton-fertilizer/yr); *N* is nitrogen content or product (%); 100 is nitrogen content unit conversion factor; *EF* is emission factor (kg-NH_3_/ton-fertilizer).

#### 3.2.2. NH_3_ Emission Calculation Method with Spatial and Temporal Resolution Enhancement Considering the Volume of Fertilizer

Method II (NH_3_ emission calculation considering the representative crops and the standard fertilizer application amount) considers the volume of fertilizer supply and the farming schedule for spatially and temporally enhancing the resolution of the NH_3_ emission calculation. The emission sources’ classification system is based on land use types (rice paddies, fields, greenhouses, and pomiculture) as shown in Table 5.

The order of the emissions calculated using Method II is shown in Figure 2.

In this study, the area of four farmland types (rice paddies, fields, greenhouses, and pomiculture) and the area cultivated for every crop type were collected [19]. Representative crops from regional farmlands were selected based on a land occupation rate of approximately 80%.

The farming schedules and standard fertilization application quantities of the selected crops were designed according to the standard cultivation method [20]. Then, weighted values for each month were calculated and the NH_3_ emission quantities for each farmland type were calculated using Equation (2). The fertilizer supply provided in Method I was assumed to have been allocated by area of each farmland type. The same values as in Table 4 were applied for emission factors.

Previous studies confirmed that over 80% of NH_3_ volatilized within 1 week in general [21,22,23,24], although temperature and seasonal effects were affected [25]. Therefore, the weighted values were calculated based on the assumption that volatilization and fertilizer application occur simultaneously.
(2)E=A×Weight×N/100×EF
where *E* is NH_3_ Emission (kg-NH_3_/month); *A* is supply volume of fertilizer (ton-fertilizer/month); *Weight* is representative crops’ weighted values for each month; *N* is nitrogen content or product (%); 100 is nitrogen content unit conversion factor; *EF* is emission factor (kg-NH_3_/ton-fertilizer).

#### 3.2.3. NH_3_ Emission Calculation Method with Spatial and Temporal Resolution Enhancement Considering Nitrogen Application

Method III (NH_3_ emission calculation considering crop-specific nitrogen application) considers the nitrogen quantities in the farming schedules for spatially and temporally enhancing the resolution of the NH_3_ emission calculation. The emission sources’ classification system is based on land use types (rice paddies, fields, greenhouses, and pomiculture) as shown in Table 6.

The order of the emissions calculated using Method III is shown in Figure 3.

Similar to Methods II and III, the spatially and temporally enhanced resolution of the considered emissions took into account the standard fertilizer application amount for farmland area based on the nitrogen content. Based on the farmland area for each selected crop type, the standard fertilizer application amount was multiplied to generate the monthly nitrogen application amount as shown in Equation (3).
(3)$$E=Cultivation\;area\times Standard\;fertilizaiton\;applicaion\;quantity\times 10^{4}\times 10^{-6}$$
where *E* is total amount of nitrogen application (kg/month); *cultivation area* is the area cultivated for every crop type (ha); *standard fertilization application quantity* is standard fertilization application quantities of the selected crops (g-N/m^2^); 10^4^ is area unit conversion factor (m^2^ to ha); 10^−3^ is fertilization unit conversion factor (g to kg).

## 4. Results and Discussions

### 4.1. Results of NH_3_ Emission Calculation Considering the Volume of Fertilizer Supply

Using Method I (NH_3_ emission calculation considering the volume of supply for each fertilizer type), the total NH_3_ emissions in 2017 in Jeolla Province was calculated to be 2,420,292 kg, as shown in Table 7. Among the 14 administrative districts in Jeolla Province, Gimje city generated the highest NH_3_ emission at 337,283 kg, followed by Iksan city (316,961 kg) and Gunsan city (313,683 kg). Together, four cities (Gimje, Iksan, Gunsan, and Jeongeup) accounted for 51% (1,231,034 kg) of the total emissions.

The monthly emission in Jeolla Province calculated using Method I was the same from March to October, as shown in Figure 4. Urea was the most frequently applied fertilizer, followed by compound fertilizers and ammonium sulfate. Currently, South Korea’s classification system identifies “farmland” as an emission source, which is not further classified into different land use types. Therefore, the spatial and temporal emission characteristics are unidentifiable.

### 4.2. Results of NH_3_ Emission Calculation Considering the Regional Representative Crops and Their Standard Fertilizer Application Amount

Using Method II (NH_3_ emission calculation considering the representative crops and standard fertilizer application amount), the total NH_3_ emission in Jeolla Province was calculated to be 2,439,895 kg, as shown in Table 8. Gimje city generated the largest NH_3_ emission at 339,241 kg, followed by Iksan and Gunsan city.

The monthly emissions varied while using Method II for the calculation, as shown in Figure 5. The monthly emission varied according to the fertilizer type when the representative crops were equal, while the overall pattern remained similar. Urea application generated the highest amount of NH_3_ emissions, followed by compound fertilizers and ammonium sulfate. Ammonium sulfate generated the lowest amount of NH_3_ emissions in all farmlands. Rice paddies fertilized by urea generated the largest emissions in May (531,000 kg), followed by fields (urea, 153,000 kg in February and October, lowest from December to January), greenhouses (highest emissions in March, even throughout the year), and pomiculture (highest emissions in February and March).

In more detail, rice is cultivated in rice paddies with an area of 98% or more, and rice is fertilized in May, July, and August under the farming schedules, so it can be seen that emissions stand out during the month. In fields, 35% of the total area of the fields is cultivated with barley, wheat, and onions, and these crops are fertilized in February and October, indicating that the emissions are high during the month. In the facility, watermelon cultivation area occupies 25%, accounting for the largest proportion, and it can be seen that the largest amount of discharge occurs because additional fertilization is given in March. In addition, in the pomiculture, 39% of the total area is cultivated for apples and persimmons, and these crops are basal fertilized in February and March. Based on South Korea’s current NH_3_ emission calculation method, spatial and temporal emission characteristics are unidentifiable. Therefore, the representative crop and farmland type identifications used in Method II will enhance NH_3_ emission controls.

### 4.3. Results of NH_3_ Emission Calculation Considering the Amount of Nitrogen Application by Crop Type

Table 9 shows the total amount of nitrogen application (15,488,710 kg) and NH_3_ emissions in Jeolla Province calculated by Method III (NH_3_ emission calculation considering crop-specific nitrogen application). The regional nitrogen application amount was converted to NH_3_ emissions by multiplying the regional nitrogen application amount by the existing fertilizer type-dependent emission factors. The emission factors were additionally weighted based on the regional supply volume of different fertilizer types. A total of 1,505,393 kg of NH_3_ were emitted. Gimje city generated the largest amount of NH_3_ emissions at 246,627 kg, followed by Iksan and Jeongeup city. The regional emission value derived from Method III differed from that derived from Method II, which is likely caused by the difference in the manner of fertilizer application between Jeongeup and Gunsan city.

As shown in Figure 6, monthly NH_3_ emissions varied with application in Method III, since the farming schedule for each crop was considered. The fertilization application schedule was the same for all crops, which resulted in a development similar to Method II. However, the results showed different emission levels. In May, July, and August, when emissions are highest, the NH_3_ emissions calculated by Method III (1,050,661 kg) were lower than those calculated with Method II (1,063,984 kg) by 13,322 kg. Method III yielded lower emissions for other farmland types as well. This confirms that actual farmlands are fertilized beyond their standard amounts set by South Korea.

### 4.4. Comparison of the Fertilized Farmland NH_3_ Emission Calculation Methods

A comparison between the previous and the enhanced NH_3_ emission calculation methods is shown in Figure 7. Method I is based on the volume of fertilizer supply, Method II has the additional considerations of regional representative crops and the standard fertilization application amount for each crop, and Method III incorporates the nitrogen application amounts, resulting in 2420, 2439, and 1505 tons of calculated NH_3_ emissions, respectively. Methods I and II, which used data based on current activities, resulted in similar emission values. Method II has the advantage that fertilizer supply data are available from the official national emission calculation. For actual compound fertilizers, the nitrogen content varies as per the manufacturer. In this study, an average value was used instead of considering the varying nitrogen content, which may have yielded different results. Method III generated the lowest emission value, which is caused by excessive fertilization. Although chemical fertilizers increase agricultural productivity, their excessive application may impoverish the soil and negatively affect crop physiology and quality [26,27,28]. The standard fertilizer application amount for each crop is defined in South Korea. However, additional amounts are optionally applied to increase productivity. Previous studies showed that nitrogen-based fertilizers are applied 1.5–2.4 times more than the standard amount [29]. Studies that evaluate changes in soil properties by farmland type showed that for rice paddies, chemical fertilizers are mostly applied within the optimal range. However, in greenhouses and pomiculture, numerous chemical fertilizers exceed the optimal range [30,31]. The results from Method III differ from those from Method I by 915 tons, demonstrating the excessive application of fertilizer on actual farmlands.

Specific regional fertilizer application excess can be ascertained using Method III, combined with the annual data on soil quality and the state of the soil, provided by the National Institute of Agricultural Sciences in South Korea. These data may be useful for the development of soil and NH_3_ emission control policies [32]. However, this combination requires complementary measures to take into account fertilizer-specific emission factors and to convert the calculated total nitrogen application amount to NH_3_.

Monthly emissions based on fertilizer application are shown in Figure 8. The regional crop characteristics reflected in Methods II and III are expected to improve the evenly distributed emissions in Method I. In Methods II and III, the largest emissions were generated in May, July, and August. This was likely caused by rice paddy NH_3_ emissions that exceeded other farmland type NH_3_ emissions. In fact, rice paddies account for the largest proportion of the area by farmland in Jeolla-do, with 68.1% of rice paddies, 22.9% of fields, 3.1% of facilities, and 5.9% of fruit trees. Rice is cultivated on an area of more than 98% in rice paddies, and according to the standard cultivation method, rice is fertilized in May, July, and August, so it is judged that the graph shown in the figure below is shown.

## 5. Conclusions

With the increase in the number of days with a high fine dust concentration and as NH_3_ is the main source of secondary PM-2.5 formation, controlling NH_3_ emissions is crucial. The US and the EU members monitor their country-specific farmland types, climate zones, precipitation values, temperatures, soil pH, and CECs, given that the main source of NH_3_ emission is the agricultural sector. In contrast, South Korea’s NH_3_ emission calculation method only considers the regional fertilizer supply volume and the nitrogen content of each product. The calculated amount of NH_3_ emission is distributed evenly for each month, which fails to reflect spatial and temporal characteristics of the crops’ cultivation periods that vary by latitude and greenhouse crops that are cultivated in winter.

In this study, the current NH_3_ emission calculation method was reviewed and two enhanced methods were derived to increase the accuracy and reliability of the results. Consequently, Methods II and III resulted in varied monthly emissions based on land use or cultivated crop types. Additionally, actual farmlands may be applying fertilizers in excess, as determined by comparing Methods I and III.

Methods II and III distinguish farmlands by land use type and consider crop-specific farming schedules by selecting representative regional crops, allowing for the identification of monthly and farmland-specific emission characteristics. In addition, the theoretical crop-specific nitrogen demand can be calculated using Method III, which may increase the precision of the NH_3_ emission calculation and the accuracy in emissions control when combined with regional data on soil quality and the state of the soil. This calculation method may be used for precise emissions control in the US or EU. Systematic data collection on future crop management plans, fertilizer application methods, and regional distributions of crops will enable precise emissions control in South Korea. NH_3_ emission prediction models can be developed in the future, with the continuous accumulation of data on soil quality and the state of the soil, climate zones, and climatic conditions.

## Figures and Tables

**Figure 1 ijerph-18-11551-f001:**
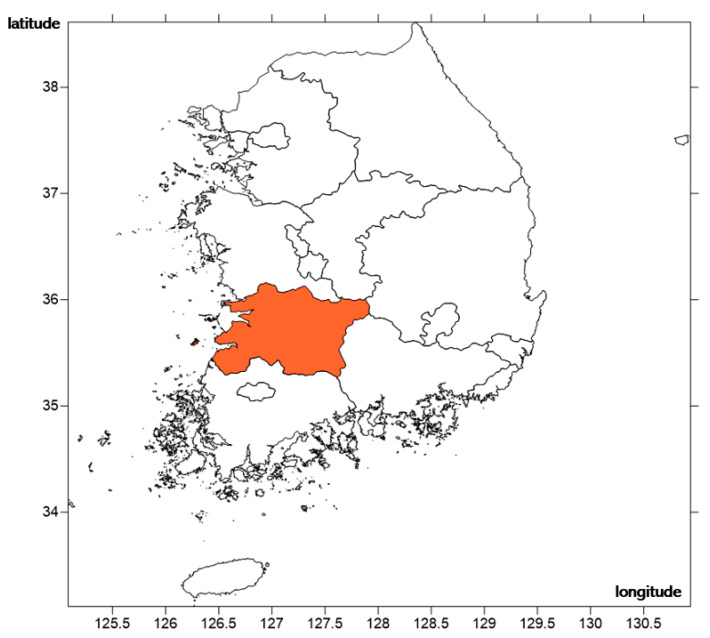
Location of study site (Jeolla Province, South Korea).

**Figure 2 ijerph-18-11551-f002:**
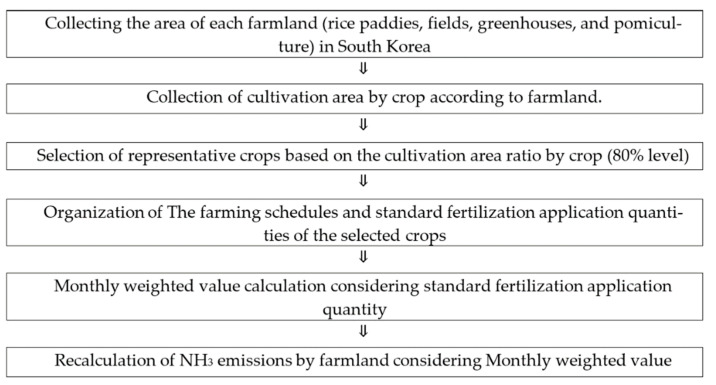
Spatial-Temporal resolution improvement work flow-chart (Method II).

**Figure 3 ijerph-18-11551-f003:**
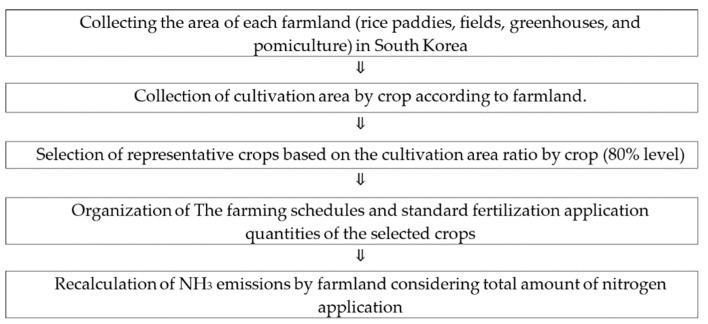
Spatial-Temporal resolution improvement work flow-chart (Method III).

**Figure 4 ijerph-18-11551-f004:**
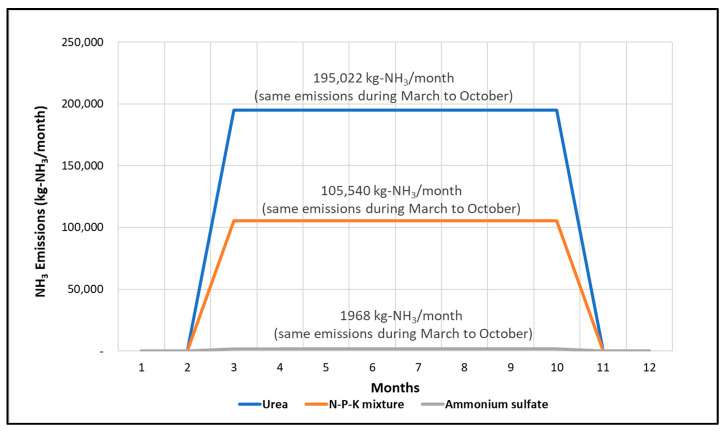
NH3 emissions by fertilizer in Jeolla-do (2017) with Method I.

**Figure 5 ijerph-18-11551-f005:**
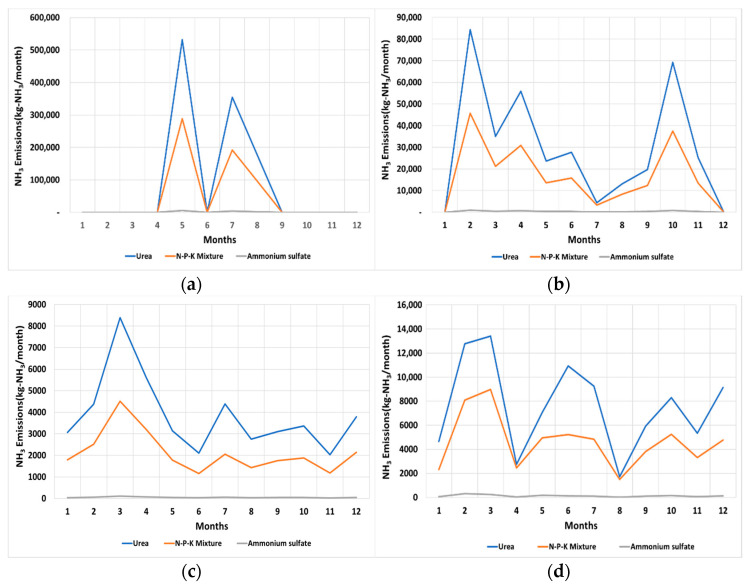
NH3 emissions by farmland in Jeolla-do (2017) with Method II. (**a**) Rice paddies; (**b**) fields; (**c**) greenhouses; (**d**) pomiculture.

**Figure 6 ijerph-18-11551-f006:**
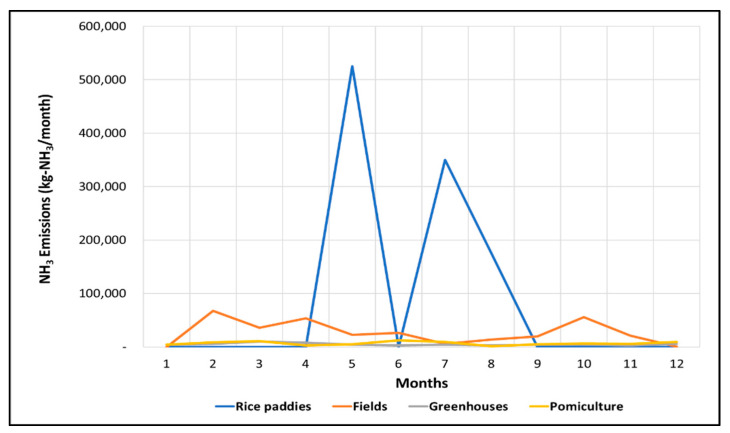
NH_3_ emissions by farmland in Jeolla-do (2017) with Method III.

**Figure 7 ijerph-18-11551-f007:**
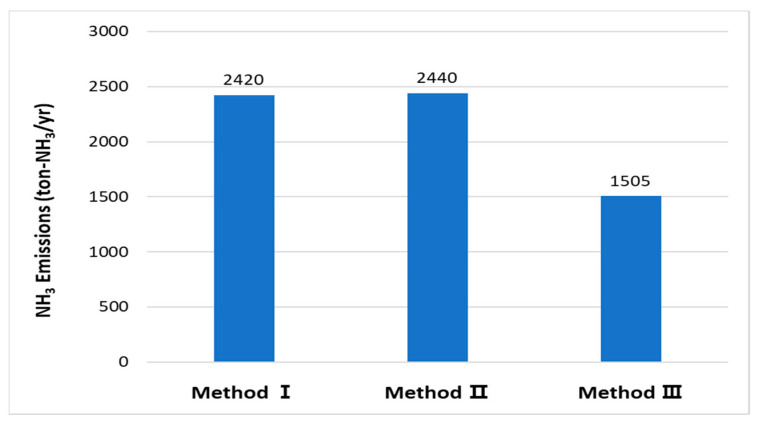
Comparison of NH_3_ emissions in fertilizer application by calculation method.

**Figure 8 ijerph-18-11551-f008:**
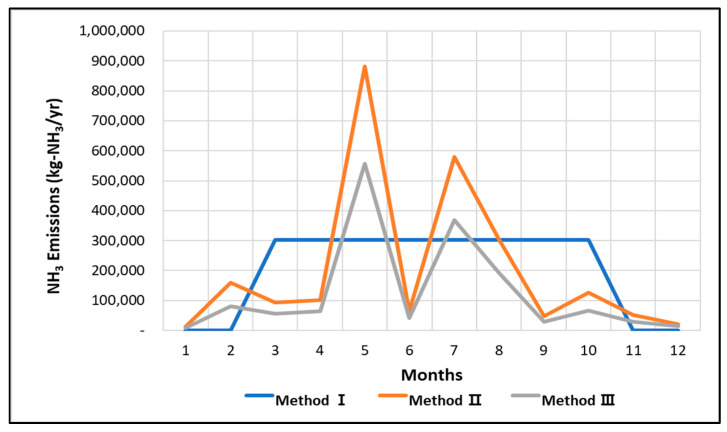
Monthly NH_3_ emissions from fertilizer application by calculation method.

**Table 1 ijerph-18-11551-t001:** Comparison and Implications of Ammonia Inventory in the World Agricultural Sector.

Classification		Content
Korea	Emission source classification	Total 10 types of fertilizer data construction status
	Estimation of NH_3_ emissions	Calculated using data on fertilizer usage taking into account nitrogen content
	Spatial-Temporal resolution	- Equal distribution of ammonia emissions from March to October - Distributed by eup, myeon, dong based on farmland area
USA	Emission source classification	Data construction status for 14 fertilizers
	Estimation of NH_3_ emissions	Using CMAQ FEST-C model for fertilizer emission scenario
	Spatial-Temporal resolution	Calculate and distribute crop cultivation across the United States
Europe	Emission source classification	Data construction status for 11 fertilizers
	Estimation of NH_3_ emissions	The calculation method is divided into Tier 1, 2, and 3, and is calculated according to the circumstances of each country - Tier 1: Utilization of basic emission factor and annual fertilizer supply - Tier 2: Considering soil pH, climatic conditions, and nitrogen loss by fertilizer - Tier 3: Using a model considering soil pH, fertilizer amount, rainfall, and temperature, etc.
	Spatial-Temporal resolution	Allocation in consideration of spatial distribution by crop type and average nitrogen input data by crop

**Table 2 ijerph-18-11551-t002:** Overview of methods for calculating NH_3_ emissions in fertilized application sector.

Classification	Content
NH_3_ emission calculation method I	- NH_3_ Emission Calculation Method Considering the Volume of Fertilizer supply - use of fertilizer supply volume, Nitrogen content of fertilizer. emission factor of fertilizer
NH_3_ emission calculation method II	- NH_3_ Emission Calculation Method with Spatial and Temporal Resolution Enhancement Considering the Volume of Fertilizer - use of fertilizer supply volume, the crops cultivated by region, the area cultivated for every crop type, the standard fertilizer application amount for each crop type
NH_3_ emission calculation method III	- NH_3_ Emission Calculation Method with Spatial and Temporal Resolution Enhancement Considering Nitrogen Application - use of the crops cultivated by region, the area cultivated for every crop type, Nitrogen requirements for each crop

**Table 3 ijerph-18-11551-t003:** The classification system of fertilizer-use agricultural land emission sources used in Korea.

Fertilizer Application	Urea
	N-P-K Mixture
	UAN
	Other N

**Table 4 ijerph-18-11551-t004:** Emission factor of fertilizer-use agricultural land emission sources used in Korea.

Classification		Emission Factor (kg-NH_3_/Ton-Fertilizer)
Fertilizer Application	Urea	141.5
	N-P-K Mixture	75.2
	UAN	97.0

**Table 5 ijerph-18-11551-t005:** The classification system of fertilizer-use agricultural land emission sources considering Method II.

Fertilizer Application	Rice Paddies	Urea
		N-P-K mixture
		UAN
		Other N
	Fields	Urea
		N-P-K mixture
		UAN
		Other N
	Greenhouses	Urea
		N-P-K mixture
		UAN
		Other N
	Pomiculture	Urea
		N-P-K mixture
		UAN
		Other N

**Table 6 ijerph-18-11551-t006:** The classification system of fertilizer-use agricultural land emission sources considering Method III.

Fertilizer Application	Rice Paddies
	Fields
	Greenhouses
	Pomiculture

**Table 7 ijerph-18-11551-t007:** NH_3_ emissions in Jeolla-do (2017) with Method I.

Province		NH_3_ Emissions (kg-NH_3_/year)
Jeolla-do	Gochang	216,020
	Gunsan	313,684
	Gimje	337,483
	Namwon	148,750
	Muju	50,549
	Buan	253,206
	Sunchang	81,459
	Wanju	157,340
	Iksan	316,962
	Imsil	98,606
	Jangsu	106,811
	Jeonju	8376
	Jeongeup	262,905
	Jinan	68,143
	Total	2,420,292

**Table 8 ijerph-18-11551-t008:** NH_3_ emissions in Jeolla-do(2017) with Method II.

Province		NH_3_ Emissions (kg-NH_3_/Year)
Jeolla-do	Gochang	215,760
	Gunsan	310,703
	Gimje	339,241
	Namwon	158,031
	Muju	50,236
	Buan	252,916
	Sunchang	82,346
	Wanju	164,513
	Iksan	317,507
	Imsil	97,504
	Jangsu	107,231
	Jeonju	8441
	Jeongeup	263,134
	Jinan	72,333
	Total	2,439,895

**Table 9 ijerph-18-11551-t009:** Amount of N application and NH_3_ emissions in Jeolla-do (2017) with Method III.

Province		Amount of N Application (kg-N/Year)	NH_3_ Emissions (kg-NH_3_/Year)
Jeolla-do	Gochang	1,837,712	172,971
	Gunsan	1,316,188	118,850
	Gimje	2,415,377	246,627
	Namwon	1,208,918	105,252
	Muju	313,051	28,266
	Buan	1,597,284	164,009
	Sunchang	573,785	48,360
	Wanju	783,759	78,769
	Iksan	1,968,993	182,377
	Imsil	585,938	53,054
	Jangsu	440,615	42,394
	Jeonju	351,643	48,746
	Jeongeup	1,710,169	179,026
	Jinan	385,278	36,692
	Total	15,488,710	1,505,393

## Data Availability

Data sharing not applicable.

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
