# Peer review of "Study on Enhanced Methods for Calculating NH3 Emissions from Fertilizer Application in Agriculture Sector"

_ijerph, 2021, doi:10.3390/ijerph182111551_

Round 1
Reviewer 1 Report
- The paper is within the scope of the subject matter of the journal and the presented topic could potentially be interesting for the society.
- The state of art methods in the field can be considered in more detail.
- The literature review is not very thorough. Some of the cited documents are simply mentioned (e.g. [4-8], [9-11], …) without a sufficient explanation of their contribution. It will be useful if the authors make more detailed comments.
- The main contribution and novelty of the work can be better explained.
The authors are encouraged to consider the following comments:
- Please check the capitalization in the article's
- The lettering of the axes of some figures is missing.
- Please check the numbering of the figures (two figures referred as Figure 4).
- Some typos.
Author Response
Dear Reviewer 1,
Thank you for your detailed review even though you are busy.
We have completed the revision based on your comments.
----------------------------------------------------------------------
The paper is within the scope of the subject matter of the journal and the presented topic could potentially be interesting for the society.
The state of art methods in the field can be considered in more detail.
The literature review is not very thorough. Some of the cited documents are simply mentioned (e.g. [4-8], [9-11], …) without a sufficient explanation of their contribution. It will be useful if the authors make more detailed comments.
: Overall, the citation has been checked again.
The main contribution and novelty of the work can be better explained.
The authors are encouraged to consider the following comments:
Please check the capitalization in the article's
: We revised it.
The lettering of the axes of some figures is missing.
: We revised all figures.
Please check the numbering of the figures (two figures referred as Figure 4).
Some typos.
: Overall, We checked it again.

Reviewer 2 Report
Comments and suggestions for Authors
”Study of Enhanced Methods for Calculating NH3 Emissions from Fertilizer Application in Agriculture Sector”
The subject is interesting and fall within the scope of the journal. The experimental dataset undoubtedly are useful and constitutes some scientific values. The presented manuscript deals with the current local problem.
In this study, global NH3 emission calculation methods were reviewed to devise potential enhancements for an accurate and a reliable method of NH3 emission calculation in the South Korean agricultural sector.
- The presented research methods need to be supplemented. There is no information on the use of natural and organic fertilizers. The volatilization of NH3 from mineral fertilizers depends on the reaction of the soil and the sorption capacity of the soil.
- For equations 1 and 2, the EF values should be given. Where did the unit (kg / ton) come from, if that is the emission factor.
- The data in the tables and the text of the manuscript should be verified. These are very high values.
- Data on NH3 emissions (kg) presented in tables 6, 7 and 8 are given per month, year or hectare? Must be completed.
- The discussion section is missing. It is very poor in subsection 4.4.
The following points may be addressed by the Authors to enhance the usefulness of the paper.
Remarks
Line 10 - it should be PM-2.5
Line 65 - to remove
Line 158 - it should be nitrogen
Line 184 - it should be nitrogen
Line 205 - it should be Results
Line 305 - - it should be Conclusions
The numbering of tables and figures should be corrected.
It is imperative to improve the readability of all figures.
In Figure 4, there is no description of the OX and OY axes
The description of the OX axis is missing in all figures.
Throughout the manuscript, the digital data records should be corrected (there should be a period instead of a comma).
The entire manuscript must be adapted to the publishing requirements.
Author Response
Dear Reviewer 2,
Thank you for your detailed review even though you are busy.
We have completed the revision based on your comments.
------------------------------------------------------------------------
”Study of Enhanced Methods for Calculating NH3 Emissions from Fertilizer Application in Agriculture Sector”
The subject is interesting and fall within the scope of the journal. The experimental dataset undoubtedly are useful and constitutes some scientific values. The presented manuscript deals with the current local problem.
In this study, global NH3 emission calculation methods were reviewed to devise potential enhancements for an accurate and a reliable method of NH3 emission calculation in the South Korean agricultural sector.
The presented research methods need to be supplemented. There is no information on the use of natural and organic fertilizers. The volatilization of NH3 from mineral fertilizers depends on the reaction of the soil and the sorption capacity of the soil.
For equations 1 and 2, the EF values should be given. Where did the unit (kg / ton) come from, if that is the emission factor.
: The emission coefficient was added to Table 4, and mentioned in Lines 159, 187-188. In addition, the units for the equation have been rewritten.
The data in the tables and the text of the manuscript should be verified. These are very high values.
Data on NH3 emissions (kg) presented in tables 6, 7 and 8 are given per month, year or hectare? Must be completed.
: The units and years in Tables 7, 8, and 9 are listed again and this study calculated emissions from 2017 data, which are specified in line 138-139.
The discussion section is missing. It is very poor in subsection 4.4.
: Section 4 describes the results and discusion together, and 4.4 contents were added a little more(line 255-263, line 330-335).
The following points may be addressed by the Authors to enhance the usefulness of the paper.
Remarks
Line 10 - it should be PM-2.5
: We revised it.
Line 65 - to remove
: line 65 is a section comparing countries' ammonia inventory. Please check again if it is correct to remove this part.
Line 158 - it should be nitrogen
: We revised it.
Line 184 - it should be nitrogen
: We revised it.
Line 205 - it should be Results
: We revised it.
Line 305 - - it should be Conclusions
: We revised it.
The numbering of tables and figures should be corrected.
: Overall, We checked it again.
It is imperative to improve the readability of all figures.
: Overall, We changed all figures.
In Figure 4, there is no description of the OX and OY axes
: Overall, We checked it again.
The description of the OX axis is missing in all figures.
: Overall, We checked it again.
Throughout the manuscript, the digital data records should be corrected (there should be a period instead of a comma).
: In Korea, comma are used instead of period in units of thousand(1,000), so we are not modified. Please let me know if I have to modify it.
The entire manuscript must be adapted to the publishing requirements.
: Overall, We checked it again.

Round 2
Reviewer 2 Report
Comment to the Authors
Dear Authors
The manuscript has scientific merit and quality to be published, enabling great information for the scientific community.
This manuscript is acceptable.
Best regards